# Using Electronic Reminders to Improve Human Papillomavirus (HPV) Vaccinations among Primary Care Patients

**DOI:** 10.3390/vaccines11040872

**Published:** 2023-04-20

**Authors:** Kathleen Hanley, Tong Han Chung, Linh K. Nguyen, Tochi Amadi, Sandra Stansberry, Robert J. Yetman, Lewis E. Foxhall, Rosalind Bello, Talhatou Diallo, Yen-Chi L. Le

**Affiliations:** 1Department of Healthcare Transformation Initiatives, McGovern Medical School, The University of Texas Health Science Center at Houston, Houston, TX 77030, USA; tong.han.chung@uth.tmc.edu (T.H.C.); linh.nguyen@uth.tmc.edu (L.K.N.); sandra.a.stansberry@uth.tmc.edu (S.S.); talhatou.diallo@uth.tmc.edu (T.D.); yen-chi.l.le@uth.tmc.edu (Y.-C.L.L.); 2Department of Pediatrics, McGovern Medical School, The University of Texas Health Science Center at Houston, Houston, TX 77030, USA; robert.j.yetman@uth.tmc.edu; 3Office of Health Policy, The University of Texas MD Anderson Cancer Center, Houston, TX 77030, USA; lfoxhall@mdanderson.org (L.E.F.); rsbello@mdanderson.org (R.B.); 4The HPV Vaccination Initiative, The University of Texas MD Anderson Cancer Center, Houston, TX 77030, USA

**Keywords:** human papillomavirus, vaccine, immunization, patient, electronic reminders, provider recommendation, patient portal, electronic health record system

## Abstract

The COVID-19 pandemic led to delays in routine preventative primary care and declines in HPV immunization rates. Providers and healthcare organizations needed to explore new ways to engage individuals to resume preventive care behaviors. Thus, we evaluated the effectiveness of using customized electronic reminders with provider recommendations for HPV vaccination to increase HPV vaccinations among adolescents and young adults, ages 9–25. Using stratified randomization, participants were divided into two groups: usual care (control) (N = 3703) and intervention (N = 3705). The control group received usual care including in-person provider recommendations, visual reminders in exam waiting rooms, bundling of vaccinations, and phone call reminders. The intervention group received usual care and an electronic reminder (SMS, email or patient portal message) at least once, and up to three times (spaced at an interval of 1 reminder per month). The intervention group had a 17% statistically significantly higher odds of uptake of additional HPV vaccinations than the usual care group (Adjusted Odds Ratio: 1.17, 95% CI: 1.01–1.36). This work supports previous findings that electronic reminders are effective at increasing immunizations and potentially decreasing healthcare costs for the treatment of HPV-related cancers.

## 1. Introduction

Over 90% of HPV-related cancers are preventable; increasing HPV vaccination rates help reduce high-risk HPV infections that lead to pre-cancers, cervical cancer, anogenital cancer, oropharyngeal cancer, and genital warts [1,2,3,4]. The American Cancer Society (ACS) and American Academy of Pediatrics (AAP) state HPV Vaccination is recommended for ages 9 through 26 and can also be recommended for patients 27–45 with shared clinical decision-making [5,6]. Patients starting their HPV vaccine before age 15 are recommended to have two doses, while patients 15 or older are recommended to have three doses [7].

The COVID-19 pandemic led to public health challenges such as delays in preventative care and decreases in immunization rates [8,9,10,11]. With the reduction in adolescents receiving vaccinations due to the COVID-19 pandemic, providers have been called on to promote vaccinations [12]. HPV vaccine administration in the US decreased by a median of 63.6% and 71.3% for ages 9–12 and ages 13–17, respectively, from 2018/2019 to March–May 2020 [11]. While vaccine administration started to increase June–September 2020 nearly to pre-COVID-19 levels in most U.S. regions, it was not enough to reach catch-up coverage for the many months of missed vaccinations. Healthcare providers were called to assess the vaccine status of their patients in order to reach herd immunity and prevent outbreaks [11]. Furthermore, vaccine hesitancy has drastically increased as well as the spread of misinformation on HPV, influenza and COVID-19 vaccines [13]. Unfortunately, exposure to negative information on HPV vaccination and mistrust in the medical system/healthcare providers is associated with parental hesitancy towards HPV vaccination [13]. Adolescents of HPV vaccine-hesitant parents were less likely to receive or initiate their HPV vaccinations [13,14]. Declines in vaccination coverage created the need for innovative interventions to increase vaccine uptake among individuals eligible for HPV vaccination. HPV vaccination coverage goals set by Healthy People 2030 are 80% [15]; however, in Texas, HPV vaccine initiation rates for ages 13–17 were 71.3% and vaccine completion rates were 51.5% in 2021 [16]. Compared to all 50 states and Washington DC, Texas ranked 48th for HPV vaccine completion rate and 44th for initiation rates [17]. Texas has experienced vaccination challenges, with the 2021 vaccine completion rate dropping 3.4% from 2020 to the 2021 rate [16]. In Harris County, for adolescents ages 9–12, initiation rates were 23.7% and completion rates were 8.2% [16]. For adolescents ages 13–17, HPV initiation rates were 67.8% and completion rates were 40.2% [16].

Systematic reviews have identified physician recommendation as the most significant predictor of vaccination uptake [18,19,20,21,22,23]. Furthermore, the Centers for Disease Control (CDC) recognizes physician recommendation as the top reason for vaccination [24]. Physician recommendation usually occurs during in-person well child or preventive care visits. However, with the ongoing pandemic, there was a dramatic decrease in healthcare utilization [8]. Innovative approaches in primary prevention were necessary to prevent further declines in vaccination. Past research identified several evidence-based strategies for improving vaccination rates. A Cochrane review found support for text messages, telephone calls, letters, postcards, and auto-dialer messages in improving immunization rates [25,26]. Based on the Cochrane review, there is strong evidence that patient reminder or recall system interventions improve vaccinations for children and adolescents [25]. At the time of our intervention development, past studies demonstrated text messaging as an effective reminder system for HPV vaccination [27,28,29,30,31,32]; however, studies on patient portal messaging for HPV reminders are limited with mixed results. One recent randomized controlled study using patient portal reminders found patient portal messaging improved influenza vaccination [33]. Whereas another study using Epic MyChart patient portal messaging for late-season Influenza vaccination found no significant differences in vaccination uptake between the intervention and control groups [23]. Another recent study found no effect of tailored reminders (based on demographics with behavior economic messaging) sent by a health system’s patient portal on influenza vaccination rates and a minimal effect (4.5%) among the young adult patient group who opened the pre-commitment portal messages asking if they planned to get vaccinated [34].

Brief educational messages can be extremely effective when structured with a strong physician recommendation highlighting the importance and the prevention of cancer [18]. We created customized electronic reminder messages that pair strong provider recommendations with brief education and emphasis on the importance and cancer prevention, a recommended strategy for strengthening provider communication about HPV vaccination [18]. To our knowledge, this is the first study to include a patient’s specific provider’s name as part of the electronic patient reminder, rather than the organization’s name [34].

The primary goal of this project was to estimate the effectiveness of customized electronic reminders (SMS, email, and patient portal message) with provider recommendations for HPV vaccination on increasing HPV vaccination rates among adolescents and young adult patients. The intervention was expected to increase appointments scheduled, clinic visits, and receipt of an HPV vaccine dose. Study results will provide useful information to healthcare institution decision-makers for utilizing electronic reminders via the electronic health record (EHR) system.

## 2. Materials and Methods

### 2.1. Study Design, Setting, and Ethics Approval and Consent

This study was approved as a Quality Improvement (QI) Project by the UTHealth Quality Improvement Project Registry (No. 2021-1135). This was a pragmatic randomized controlled study at 13 multiple primary care clinics of one organization in Houston, TX. Since this is a QI study, patient consent was waived. All patients received usual care whereas the intervention group received customized electronic reminders in addition to usual care.

### 2.2. Study Population

The study population included 7408 patients age 9–25 who: (1) had at least one office visit at a primary care (family medicine, general medicine, pediatrics) clinic from 1 January 2021 to 31 December 2021, (2) were eligible to receive the HPV vaccine, and (3) had a valid communication method such as mobile phone number, email address, or patient portal activation documented within the EHR system. Our EHR system is Epic; MyChart is a web-based patient portal in Epic that allows patients access to their medical records and to communicate with their providers. Eligibility included patients who had not yet received an HPV vaccine (or had no record of vaccination in their patient record) as well as patients who initiated but did not complete their HPV vaccine series. Vaccine series completion was defined based on the CDC HPV vaccination recommendations [7]. Patients were excluded if they were pregnant, had immunization contraindications in their medical record, or had documented vaccine refusal.

### 2.3. Randomization

Patients were randomly assigned by stratified randomization into two groups, usual care (*n* = 3703) or intervention (*n* = 3705), based on age, vaccine status, sex and clinic location using Stata 14.0 (StataCorp., College Station, TX, USA).

### 2.4. Study Intervention and Usual Care Group

The intervention group received one type of electronic reminder, following a hierarchical order: (1) SMS if the patient opted in to receive SMS; (2) a patient portal message via MyChart if the patient opted in for MyChart messages but not SMS; or (3) an email reminder if the patient did not opt in to SMS or MyChart messages. The electronic reminder was addressed to the parent/caregiver or patient (based on the patient’s age) and included physician recommendation and brief education (See Table 1). Messages to patients under 18 years of age were addressed to the parent/caregiver of the patient. Messages to patients 18+ years of age were addressed directly to the patient.

The control group received usual care which included in-person provider recommendations, visual reminders in exam waiting rooms, bundling of vaccinations, and phone call reminders. Patient appointments, clinic visits, and HPV vaccine status were captured in the patient’s EHR. For ease, we will refer to the control group as the usual care group.

### 2.5. Intervention Period

We used Epic’s Campaigns application, a mass patient outreach tool within Epic to send messages (SMS, patient portal, email) to the intervention group. Epic Campaigns allows users to send patient portal messages and emails through the EHR platform. However, SMS messages were sent using a data file exported from Epic and uploaded into a third-party SMS system, Tavoca. Messages were sent once a month for up to three months, based on whether a patient took “action” after the message. Patients who scheduled a future appointment or nurse visit or completed their HPV vaccination series were counted as a success and did not receive future messages. However, patients who did not take “action” received monthly reminders for an additional two months. All patients who opted out of electronic reminders were removed from receiving additional messages. We followed patients for 6 months after the intervention start, to allow time for patients to schedule and complete their HPV vaccine (Figure 1).

### 2.6. Study Measures and Data collection

Patient demographic and clinical data, including date of birth, gender, vaccine dose, vaccine status, appointment scheduling, and clinic visits were extracted from the EHR system to determine the eligible population, baseline data, and follow-up data.

The study outcomes include HPV vaccination-associated appointments, clinic visits, and vaccination rates. We considered patients making future appointments as a proxy for vaccine intention. Based on the Theory of Planned Behavior (TPB) and past research, vaccination intentions can predict behavior [35,36]. Therefore, we compared rates of appointment scheduling and clinic visit(s) between the two groups. We included appointments and visits for well child, well women exam, nurse visit or physical only and excluded sick visits.

All additional HPV vaccinations during the intervention period were estimated among all patients. The vaccine initiation rate was also calculated as the percentage of eligible patients who had ≥1 dose but had not yet completed the series among eligible patients who had not received the HPV vaccine. The vaccine completion rate was calculated as the percentage of patients who completed the vaccine series among the entirety of eligible patients.

### 2.7. Analysis

Descriptive analysis was performed with frequency distributions. A chi-square test was performed for categorical variables. We applied multiple logistic regression models for analyses of the intervention effects. Effect sizes were presented as odds ratios with 95% confidence intervals. Socioeconomic factors and the three stratifying variables were adjusted as potential confounders (age, sex, race/ethnicity, insurance and vaccine status). An intent-to-treat analysis was used as a primary analysis. There were differences between the randomized intervention group and the actual intervention group who received the intervention mainly due to the patients’ opt-out status of all the modes of the intervention (SMS, email and patient portal) and patients’ incorrect phone or email information which make them unreachable. Therefore, we also conducted a secondary analysis among the intervention group who eventually received the intervention in comparison with the usual care group in order to assess the practical effectiveness of the intervention. The statistical analysis was conducted using Stata 14.0 (StataCorp., College Station, TX, USA).

## 3. Results

### 3.1. Patient Demographics

Table 2 displays patient demographics at baseline. A total of 7408 patients who met eligibility were randomized. The intervention group and usual care group followed nearly identical distributions in age, sex, vaccination status, race/ethnicity and insurance type. Patients ages 9–14 accounted for approximately 40%, ages 15–18 accounted for 19–20% and ages 19–25 accounted for 40% of the population. Approximately 55% of patients were female and 68% were not yet vaccinated. Non-Hispanic White, non-Hispanic Black, Hispanic and other races/ethnicity each accounted for 25% of the population. Privately insured patients accounted for 55%, Medicaid 36%, and uninsured 8–9% of the population for both groups (intervention and usual care).

### 3.2. Appointment Scheduling and Clinic Visit

The intervention group had higher rate of appointment scheduling for HPV vaccination than the usual care group (Adjusted Odds Ratio (AOR): 1.12, 95% CI: 1.00–1.26). The intervention group also had a higher rate of clinic visits than the usual care group (AOR: 1.12, 95% CI: 0.90–1.38). However, the differences in rates were not statistically significant (Table 3).

### 3.3. HPV Vaccination Rates

The intervention group showed a 17% statistically significant increase in the odds of obtaining an additional HPV vaccination than the usual care (AOR: 1.17, 95% CI: 1.01–1.36) (Table 3). Breaking down the analysis by initiation and completion rate, differences in vaccine initiation rate (AOR: 1.22, 95% CI: 0.98–1.51) and completion rate (AOR: 1.12, 95% CI: 0.90–1.38) were noted between the intervention and usual care group but were not statistically significant (Table 4).

#### Secondary Analyses

Of the 3705 patients in the intervention group, only 2768 received one or more electronic reminders. The distributions of age, sex, vaccine status, race/ethnicity and insurance between the two groups were slightly different. The usual care group had a higher proportion of patients ages 9–14 (41.5%) compared with the intervention group (33.6%) and a slightly higher proportion of patients under Medicaid (36%) compared with the intervention group (32.4%) (Appendix A).

Patients who received the intervention were more likely to schedule an appointment (AOR: 1.2, 95% CI: 1.06–1.37) and have an additional HPV vaccination (AOR: 1.35, 95% CI: 1.14–1.59) compared to the usual care group patients. In addition, among the intervention group HPV vaccine initiation rate (AOR: 1.32, 95% CI: 1.04–1.68) and completion rate (AOR: 1.32, 95% CI: 1.12–1.56) were significantly higher than those among the usual care group. (Appendix A).

## 4. Discussion

Our study assessed how customized electronic reminders (SMS, email, and MyChart) that included a provider recommendation and education on cancer prevention, impacted HPV vaccination rates among adolescents and young adults in a primary care setting. The intervention was essential to impacting the drop in HPV vaccinations due to the COVID-19 pandemic. Our study had several strengths. We utilized a practical population-based approach to increase vaccination rates in an outpatient, primary care setting. We had a relatively large sample size. Our organization also has bidirectional data exchange with two external immunization information systems (ImmTrac2 and CareEverywhere). This allowed our patient immunization records to contain both immunizations administered by our organization and elsewhere. We also leveraged the EHR to implement a technology-based intervention. Although the initial cost of adopting a systems-level patient reminder tool can be significant, maintenance costs are relatively minimal. In addition, adopting automated patient reminder tools may also minimize the human capital resources that are typically needed to manually place individual phone calls and allow for the customization of messages and tailored interventions. To the best of our knowledge, our study is one of the first to use customized patient portal messages for HPV vaccination [37], in addition to using SMS and emails.

The latest Cochrane review of patient reminder and recall systems to improve immunization found a “high certainty of evidence” to improve immunizations through single-method reminders: postcards, text messages, auto-dialer phone calls and moderate evidence for telephone [26] calls and letters. At the time of Jacobson et al., there were no studies utilizing patient portal messages that met the Cochrane review inclusion criteria. Previous research has found internet and mobile technology (such as recall prompts, SMS, interactive videos, phone calls, and email) effective at improving HPV vaccination and completion [37]. In reviews by Francis et al. (2017) and Acampora et al. (2020) [37,38], the EHR was used for provider recall interventions rather than patient reminder messages. As previous researchers have suggested exploring [26], we utilized an EHR patient portal system and its secure messaging in our intervention and found that the intervention group was 17% more likely to receive an additional HPV dose than the usual care group. Our study also provides evidence that customized electronic patient reminders tailored to include an HPV Vaccine recommendation from the patient’s primary care provider may increase HPV vaccination. The literature supports provider recommendations to increase vaccination, and this study shows an electronic practical solution for healthcare clinics.

Another strength is that our study is one of a few studies that assess future preventive care appointments scheduled during the intervention/post-intervention period as one of the study outcomes. Future scheduled appointments may function as a process measure for measuring intervention effectiveness. Scheduled appointments and clinic visits might serve as an intermediate outcome of the study impact prior to the completion of a vaccination, which would be the primary outcome. For example, a patient or caregiver who received the HPV reminder may have scheduled an appointment for 7 months from the time they received the message with the intention of receiving the HPV vaccine in the future. This patient may have the intention to obtain the HPV vaccine but would not be included among those who received an HPV dose during the study period. In a randomized trial by a managed care organization (MCO), an immunization reminder/recall intervention on routine childhood immunizations (Meningococcus, pertussis, HPV) had a modest impact on preventative care visits (mail intervention: 65% visit, telephone 63% visit, and control group: 59% visit) [39]. Similarly, our study did not find statistically significant differences in appointment scheduling and clinic visits between the usual care and intervention groups, rates in scheduling (20.3% vs. 18.9%) and clinic visits (11.42% vs. 11.32%) were higher among the intervention group compared to the usual care group.

Past studies that have small effect sizes between the intervention and usual care group may be attributed to high standards for usual care [26,40]. In our community-based clinics, usual care typically involves one or more of the following evidence-based strategies: in-person provider recommendation, visual reminders in exam waiting rooms, bundling of vaccinations, and phone call reminders for patients. Some studies have found that patient navigator phone calls increased vaccine completion by 10% more than the control group [41]. A combination of patient reminder/recall and provider reminder interventions has been shown to likely improve immunizations based on moderate certainty evidence [26]. Therefore, it is possible that the additional benefit from our intervention showed a marginal impact when compared to our usual care. 

Although our vaccination rate was lower than previous HPV vaccination studies [38], small effect sizes for interventions that increase immunizations are considered clinically meaningful due to their almost universal recommendation for populations and the protection of public health [26]. Every single HPV vaccination that may prevent cancer is a win for primary care providers and public health professionals.

One potential study limitation is that our study was implemented when recommendations for HPV vaccinations had just been expanded to begin at age 9. This may have contributed to our study’s overall low initiation and completion rates. In 2018, the AAP and in 2019, the ACS recommended HPV vaccination starting at age 9 [5,6], while the Advisory Committee on Immunization Practices (ACIP) recommends starting at ages 11–12, and as early as age 9 [7,42]. Considering this more recent change there is a slower uptake in vaccination by ages 9–12, with most adolescents completing vaccination as “catch-up” shots [43]. Prior to the start of our study, most providers in our clinics were routinely recommending HPV vaccination at 11 years of age and we were not systematically outreaching or tracking HPV vaccination rates for 9–10 year olds. The ACIP recommends HPV vaccination to all persons up to age 26. Heeding these recommendations, our study included patients in the 18–25 age group. Historically, this age group has lower HPV vaccine completion rates than adolescents 13–17 [44]. Including the 18–25 age group in our target population may have also lowered our study’s overall low rates. Current data dashboards on HPV vaccinations focus on adolescents 13–17 [45]; therefore, it is difficult to estimate national and statewide changes in HPV vaccinations by age group. Technology-based interventions such as ours that rely on automated reminders can provide a wider range of outreach to patients that include the 18–26 year age range with minimal additional costs.

Another potential study limitation may be delays in receipt of immunization records or missing immunization records. This may have also contributed to our study’s lower vaccination rates. Our study took place in Texas which has an opt-in immunization registry. Patients must provide consent to have their immunizations stored in ImmTrac2 and shared with healthcare organizations. We are not able to retrieve the immunization records of patients who have not consented. Even if patients have consented to ImmTrac2, it might take time for patients’ external vaccination information to be transmitted via ImmTrac2. Another source for external immunization data is CareEverywhere. Although our patients are automatically opted in to CareEverywhere access, participation from external vaccine administering organizations is voluntary. It is possible that our study timeframe may not have allowed us adequate time to capture HPV vaccinations that our patients received elsewhere. It is also possible that some patients may have received vaccinations from external organizations that do not participate in ImmTrac2 or CareEverywhere. Lastly, the pragmatic approach used in this quality improvement study may have also contributed to the small effect size.

Another study limitation is that our study population may not be representative of the general population. Our intervention utilized electronic reminders (email, text or patient portal) which excluded patients that did not provide a way for us to contact them electronically as well as patients who may not have access to electronic communications. Future studies may need to address technology inequities if they utilize a web-based or electronic approach for their intervention.

Even with the study limitations and limited generalizability we describe above, our study makes a meaningful contribution to the literature on HPV vaccination interventions. As a department that implements systematic operations within a healthcare institution, we implemented electronic patient reminders as a QI project, randomized patients, and implemented this intervention without notice to patients and providers and with minimal disruption to existing clinic workflows. Considering our practical setting, our study might show more realistic results while preserving as much internal validity as possible compared to a typical randomized controlled study. Although our study provides valuable pragmatic results, the results may not be generalizable to other locations or populations outside of the healthcare system.

HPV-related cancers have caused a considerable economic burden. In 2020, the total annual medical cost of cervical cancer was estimated to be $2.3 billion and the average per-patient costs for medical services were estimated to be around $160,200, which is the sum of the average annualized cost for each phase of care (initial, continuing and end-of-life) [46]. Given that HPV vaccines are extremely effective in preventing HPV-related cancers [47] and very low cost compared to the total per-patient cost for treating HPV-related cancers [48], vaccinating against HPV can be highly cost-saving for an individual. Therefore, even for studies that show a small increase in HPV vaccinations, the potential healthcare cost savings for vaccinated patients are non-negligible.

Many healthcare organizations use the EHR, making use of communication technologies for interventions more feasible with little added cost compared to human capital [37]. With the novelty of using patient portal messages, particularly Epic Campaigns, to increase vaccinations, and the likelihood of more adoption of the EHR for improving patient vaccinations, future studies should conduct an economic evaluation to investigate the positive financial impact on the healthcare system over time.

Furthermore, clinical practices can apply this messaging framework and patient portal delivery method to other immunizations. As suggested by Jacobson et al. (2018) and Francis et al. (2017) [26,37], future researchers may consider using EHR and technology to customize interventions based on risk or culture.

## 5. Conclusions

Improving HPV vaccination uptake is an important public health goal requiring continued attention due to the effects of the COVID-19 Pandemic, such as decreased healthcare utilization and growing mistrust in vaccines. Our study adds to the literature that customized electronic reminders can improve HPV vaccinations among eligible individuals. Future research should investigate whether our message framework is useful to other patient populations and if this pragmatic approach is applicable to other healthcare settings. Healthcare organizations may consider customized electronic reminders for other immunizations such as flu or pneumonia as well as other preventative care activities such as wellness visits and screenings that declined during the COVID-19 pandemic but are crucial for optimal health.

## Figures and Tables

**Figure 1 vaccines-11-00872-f001:**
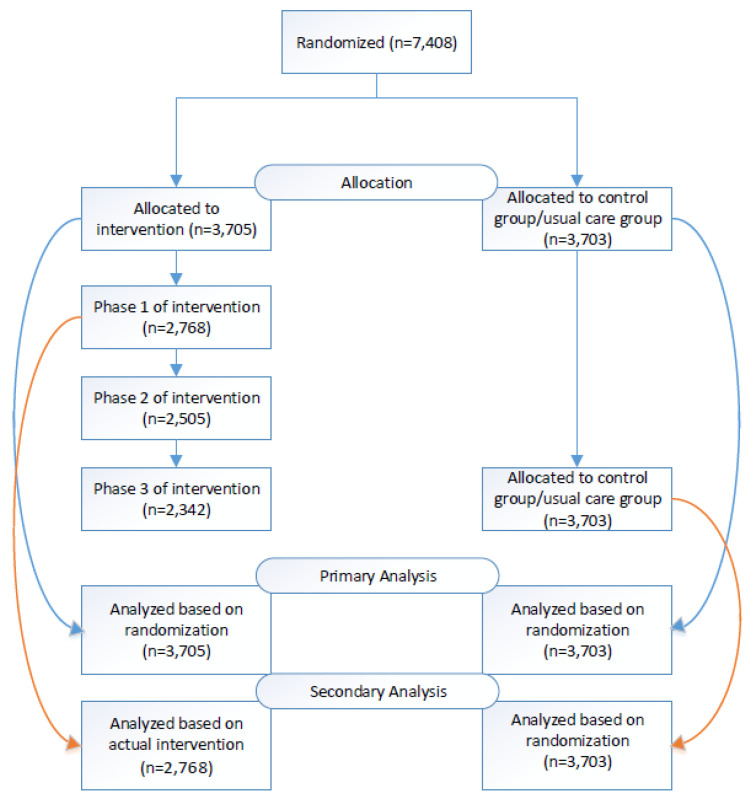
Consort flow diagram-participant randomization, intervention, and analysis.

**Table 1 vaccines-11-00872-t001:** Message framework for electronic patient reminders for HPV.

	Patient Less Than 18 Years Old	Patients 18 Years or Older
Initial Message	Dr. Yetman [PCP name] recommends Jane [Patient name] complete their HPV vaccination to prevent cancers caused by HPV. To schedule go to [Scheduling link]. Reply OK to confirm receipt.	John [Patient Name]: Dr. Foxhall [PCP name] recommends you complete your HPV vaccination to prevent cancers caused by HPV. To schedule, go to [Scheduling link]. Reply OK to confirm receipt.
Second or Third Message	Reminder: Dr. Yetman [PCP name] recommends Jane [patient name] complete their HPV vaccination. To schedule go to [Scheduling link]. Reply OK to confirm receipt.	Reminder for John [Patient Name]: Dr. Foxhall [PCP name] recommends you complete your HPV vaccination. To schedule, go to [Scheduling link]. Reply OK to confirm receipt.

**Table 2 vaccines-11-00872-t002:** Baseline Characteristics by randomized group.

Baseline Characteristics	Electronic Reminder (N = 3705) N (%)	Usual Care (N = 3703) N (%)	*p*-Value
Age			0.689
9–14	1511 (40.78)	1535 (41.45)	
15–18	721 (19.46)	693 (18.71)	
19–25	1473 (39.76)	1475 (39.83)	
Sex			0.942
Male	1676 (45.24)	1672 (45.15)	
Female	2029 (54.76)	2031 (54.85)	
Vaccine Status			0.834
Not initiated	2542 (68.61)	2549 (68.84)	
Initiated	1163 (31.39)	1154 (31.16)	
Race/Ethnicity			0.176
Non-Hispanic White	865 (23.35)	939 (25.36)	
Non-Hispanic Black	955 (25.78)	962 (25.98)	
Hispanic	911 (24.59)	874 (23.60)	
Other/Unknown	974 (26.29)	928 (25.06)	
Insurance			0.134
Medicaid	1317 (35.55)	1337 (36.11)	
Private (Managed Care)	2014 (54.36)	2049 (55.33)	
Uninsured	338 (9.12)	291 (7.86)	
Other	36 (0.97)	26 (0.70)	

**Table 3 vaccines-11-00872-t003:** Multiple logistic regression * analysis results of HPV outcomes.

	Electronic Reminder	Usual Care		
	(N = 3705)	(N = 3703)		
HPV Outcomes	N (%)	N (%)	Adjusted OR (95% CI *)	*p*-Value
Appointment Scheduling	752 (20.3)	700 (18.9)	1.12 (1.00, 1.26)	0.056
Clinic Visit	423 (11.42)	419 (11.32)	1.07 (0.94, 1.23)	0.300
All Additional HPV Vaccination(s)	450 (12.15)	402 (10.86)	1.17 (1.01, 1.36)	0.036

* Model controlled for age, sex, race/ethnicity, insurance and vaccine status.

**Table 4 vaccines-11-00872-t004:** Subgroup vaccination outcomes.

	Electronic Reminder	Usual Care		
	(N = 2529)	(N = 2538)		
	N (%)	N (%)	Adjusted OR (95% CI**)	*p*-Value
HPV vaccine initiation rate	206 (8.15)	181 (7.13)	1.22 (0.98, 1.51)	0.076
	**Electronic Reminder**	**Usual Care**		
	**(N = 3705)**	**(N = 3703)**		
	**N (%)**	**N (%)**	**Adjusted OR** **(95% CI**)**	***p*-Value**
HPV vaccine completion rate	225 (6.07)	207 (5.59)	1.12 (0.90, 1.38)	0.302

## Data Availability

The data presented in this study are available on reasonable request from the corresponding author. The data are not publicly available due to containing protected health information.

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
