# Peer review of "Using Electronic Reminders to Improve Human Papillomavirus (HPV) Vaccinations among Primary Care Patients"

_vaccines, 2023, doi:10.3390/vaccines11040872_

Round 1
Reviewer 1 Report
This paper describes a pragmatic randomized controlled study of an electronic reminder intervention to improve HPV vaccination at several primary care sites in Texas. The authors found a significant 17% increase in the odds of HPV vaccine uptake in the intervention group compared to the usual care group. Rates of appointment scheduling and clinic visits were also higher in the intervention group, though the results did not reach statistical significance. The paper highlights an effective approach that leverages EHR technology to improve immunization rates. My minor comments are below.
Abstract/title
1. The abstract and title could be improved by adding more specific information about the population involved (e.g., Texas primary care patients)
Materials/Methods
2. Lines 102-103: How many primary care clinics were involved?
3. Line 153: "The study outcomes include HPV vaccination associated appointments, clinic visits, 153 and vaccination rates." Were appointments and clinic visits only included if they were specifically for HPV vaccination? If so, how was this defined?
4. Line 168: Can you list the sociodemographic factors that were included in the regression models? I see it in the footnote of Table 3, but it would be helpful to include here in the text as well.
Tables
5. I found that the order of the columns in tables 3 and 4 and supplemental tables 2 and 3 made them a little difficult to read. I'd recommend putting the two "N (%)" columns first, followed by the "Adjusted Odds Ratio" column, and finally the "p-value" column.
Reviewer 2 Report
Dear authors,
thank you so much for your efforts in scientific research. This is a very interesting and timely article. This article presents an interesting evaluation of the effectiveness of personalized electronic HPV vaccination reminders in increasing HPV vaccination rates among adolescents and young adults after the decrease due to pandemic COVID-19. The the size of the patient groups examined certainly appears to be significant, and so does the statistical method used.
The study found that the intervention group, which received both usual care and electronic reminders, had significantly higher odds of undergoing additional HPV vaccinations than the control group, which received only usual care. In conclusion, this study highlights the importance of exploring new ways to engage people to resume preventive behaviors during periods of lower health care utilization, such as pandemic COVID-19, and the potential role of technology in improving health care outcomes.
In addition, this article turns out to be very useful for subsequent statistical meta-analysis studies, which could confirm the results obtained, or find new correlations.
Thanks again for your contribution.
Best regards.
Author Response
Thank you. We appreciate you reviewing our work.
Reviewer 3 Report
The ms ‘ Using Electronic Reminders to improve human papillomavirus 2 (HPV) vaccinations among a patient population ‘ by Hanley et al propose to improve the vaccination status using reminders by electronic gradates. Various intervention systems are used by researchers and clinicians for vaccination against HPV and the current proposed intervention can be one of them.
Following parts need modification.
Please read the following numbers as line numbers
Abstract
13: It is wrong. COVID-19 increases (over used) health care units. So mention any other major reason.
23: Use “effect size” statistics and mention the value here.
Also mention the other statistical values in this section. You can start this section by putting the current status and need of vaccination against HPV.
Introduction
Line 30: Delete
31-33: Mention some values about it status
39: Yes delay could be one of the reason due to covid-19
40: Cite one useful article as https://pubmed.ncbi.nlm.nih.gov/32339832/
43: How much decreased, that’s the main point of this study/
45: How much and what was he reason.
49: Is it negative info or wrong info?
62” Delete Current Research.
92: Delete Main Aim and principal conclusion.
93-98: Make a clear hypothesis and you can add how vaccination is important to control even pandemic COVID-19. Follow the article https://pubmed.ncbi.nlm.nih.gov/34162043/
Materials methods
Add a consort flow in this section, may be Table 1 can be converted into a consort flow.
165: Calculate effect size
Discussion
Add what are the limitations and strength of this study .
335-340: Cant is be a common strategies for all types of treatment where delay is a factor? So, conclude the study accordingly with future prospective.
Author Response
Thank you. Please see attached attachment.

Reviewer 4 Report
Thank you for inviting me to review this work entitled."Using Electronic Reminders to improve human papillomavirus 2 (HPV) vaccinations among a patient population", which aimed to evaluate the effectiveness of using customized electronic reminders with provider recommendation for HPV vaccination to increase HPV vaccinations 16 among adolescents and young adults.
In general, the research idea is nice and I believe it is important.
Minor comments were raised and should be addressed before proceeding with the decision:
1. I believe it is better to use the term "conventional" care instead of usual care.
2. You need to highlight the study's main findings comprehensively in the abstract (with numbers). Especially the number of respondents..etc
3. at the end of the abstract, highlights the clinical implication related to HPV.
4. Introduction: written nicely; thank you
5. Methods:
-Regarding "The study population included 7,408 patients ages 9-25", are these documented statistics? Please cite the reference.
-did you use a certain application for the randomization process?
Statistical analysis:
-Please clarify the dependent/independent variables.
-checking the assumptions for logistic regression, what about the simple regression? Please add its data as supplementary material.
Discussion:
Please add the study limitations at the end of this section (using a clear subheading).
